# Rapid Selection and Ordering of In-Context Demonstrations via Prompt Embedding Clustering

**Kha Pham**[1]    **Hung Le**[1]    **Man Ngo**[2]    **Truyen Tran**[1]

[1] Applied Artificial Intelligence Institute, Deakin University
[2] Faculty of Data Science in Business, Ho Chi Minh University of Banking, Vietnam

[1] {phti, thai.le, truyen.tran}@deakin.edu.au
[2] mannm@hub.edu.vn

## Abstract

While Large Language Models (LLMs) excel at in-context learning (ICL) using just a few demonstrations, their performances are sensitive to demonstration orders. The reasons behind this sensitivity remain poorly understood. In this paper, we investigate the prompt embedding space to bridge the gap between the order sensitivity of ICL with inner workings of decoder-only LLMs, uncovering the *clustering property: prompts sharing the first and last demonstrations have closer embeddings*, with first-demonstration clustering usually being stronger in practice. We explain this property through extensive theoretical analyses and empirical evidences. Our finding suggests that the positional encoding and the causal attention mask are key contributors to the clustering phenomenon. Leveraging this clustering insight, we introduce Cluster-based Search, a novel method that accelerates the selection and ordering of demonstrations in self-adaptive ICL settings. Our approach substantially decreases the time complexity from factorial to quadratic, saving 92% to nearly 100% execution time while maintaining comparable performance to exhaustive search.

## 1 Introduction

In-context Learning (ICL) is a remarkable emergent ability of Large Language Models (LLMs) to perform few-shot learning. This means they can answer new queries based on just a handful of related demonstrations, even without explicit training for the specific task (Brown et al., 2020). While highly effective, research has shown ICL performances can be extremely sensitive to demonstration orders (Zhao et al., 2021; Liu et al., 2021; Lu et al., 2022; Liu et al., 2023). To the best of our knowledge, the reasons behind this sensitivity remain unclear.

In this paper, we investigate the prompt embedding space to explain the order sensitivity of decoder-only LLMs in ICL. Specifically, we focus on the last token embedding in the final LLM layer, which is the input for the next-token prediction head. Our investigation reveals the **clustering property**: *prompts with the same first and last demonstrations form clusters in the embedding space* (Figure 1), with first-demonstration clustering usually being stronger in practice. Our finding aligns with recent research on the special roles of the first and last demonstrations in ICL (Liu et al., 2023; Janik, 2023). But unlike prior works, our clustering property links the order sensitivity of ICL and inner workings of LLMs, allowing more systematic insights into LLMs' behaviors.

We provide extensive analyses to confirm the clustering property. In particular, we visualize prompt embeddings in 2D spaces using UMAP, run K-Means clustering on high-dimensional embedding spaces, and quantify the importance of input tokens by their partial derivative norms. Experimental results consistently support the existence of clusters.

Further, we provide theoretical and empirical explanations for the clustering behaviors. Theoretically, we prove that first-demonstration clustering is associated with causal attention mask under ideal conditions. To validate this relationship in practice, we train decoder-only Transformers from scratch

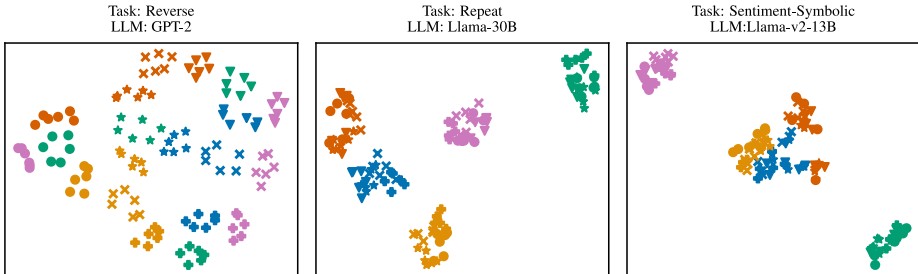

Figure 1: *Prompts are clustered by first and last demonstrations*. We consider prompts with the same query and demonstrations but in different orders. For each prompt, we show its UMAP 2D projection of the last embedding in the final LLM layer. Points with the same colors have the same first demonstrations, and points with the same shapes have the same last demonstrations. Interestingly, points with similar colors and shapes form cluster. This property holds for various LLMs and ICL tasks.

on WikiText2 dataset (Merity et al., 2017) with and without causal attention mask, achieving similar outcomes: causal attention mask contributes to the importance of beginning tokens. We also perform similar experiments with and without positional encoding, and find that the interplay of the causal structure and the positional encoding contributes to the importance of ending tokens.

Finally, we leverage the clustering property to accelerate self-adaptive approaches (Wu et al., 2022) when dealing with the costly problem of demonstration selection and ordering. Our proposed method, dubbed Cluster-based Search, substantially decreases the time complexity from the factorial order of exhaustive search to quadratic order. We apply Cluster-based Search in two selection scenarios: one with the ideal selection criterion and another with the practical entropy-based criterion (Lu et al., 2022). In both cases, our proposed method achieves competitive accuracies compared to exhaustive search while being significantly faster – saving $92\%$ to nearly $100\%$ execution time.

In short, our contributions include: (1) Establishing the clustering property by first & last demonstrations and verifying it from various angles, (2) analyzing its underlying mechanisms both theoretically and empirically, and (3) leveraging it to propose Cluster-based Search, which significantly accelerates the selection and ordering of in-context demonstrations in self-adaptive ICL settings.

## 2 PRELIMINARIES

### 2.1 TERMINOLOGIES AND NOTATIONS

**Transformer** We consider decoder-only LLMs built upon the Transformer architecture. The Transformer takes the input as a sequence of $n$ tokens $\{x_i(0)\}_{i=1}^n \in \left(\mathbb{R}^d\right)^n$. This input sequence is transformed through $T$ Transformer layers, of which the $t$-th layer ($t = 1, \ldots, T$) receives the sequence $\{x_i(t-1)\}_{i=1}^n \in \left(\mathbb{R}^d\right)^n$ as inputs and outputs another sequence $\{x_i(t)\}_{i=1}^n \in \left(\mathbb{R}^d\right)^n$. We primarily focus on the last embeddings of Transformer layers, i.e. $x_n(t)$, which we will denote as $x_{-1}(t)$ for $t < T$ and $x_{-1}$ when $t = T$. We denote $\langle x_i, x_j \rangle$ as the inner product of vectors $x_i$ and $x_j$.

**In-context learning (ICL)** An ICL prompt consists of an ordered sequence of $k$ *demonstrations* $E = (e_1, \ldots, e_k)$ and a *query* $q$. Each demonstration $e_i$ includes an input $e_i^{\text{in}}$ (e.g. a sentence) and its associated output $e_i^{\text{out}}$ (e.g. the sentence's sentiment). The query $q$ only consists of an input $q^{\text{in}}$, and the LLM will predict its associated output, i.e. $\hat{q}^{\text{out}} = \text{LLM}(E, q)$. The prediction $\hat{q}^{\text{out}}$ is compared with the ground-truth output $q^{\text{out}}$ to determine whether the LLM answers the query correctly. In Section 3 and Section 4, we investigate LLMs' behaviors on ICL prompts built upon the same query and the same set of demonstrations but in different orders.

### 2.2 TASKS

We consider two types of tasks: classification and reasoning.

For text classification, we consider tasks of *sentiment classification* and *language identification*. We use data from SST-2 (Socher et al., 2013) dataset for sentiment classification, where each demonstration is a pair of a natural language sentence and its associated sentiment. Prompts in language identification task are similar, except that each demonstration is a pair of a sentence and its associated language. Dataset for language identification is taken from HuggingFace (HuggingFace, 2021). We

also consider tasks of *symbolic sentiment classification* and *symbolic language identification*. The labels of demonstrations in symbolic sentiment classification are not "positive" or "negative", but replaced by random strings. For example, "positive" becomes "a b c" and "negative" becomes "x y z", so the model should predict "a b c" or "x y z" instead of "positive" or "negative". Similarly for symbolic language identification.

For reasoning tasks, we consider symbolic reasoning, common-sense reasoning, and mathematical arithmetic reasoning. The symbolic reasoning tasks include *reverse* and *repeat*. For reverse, the demonstration input is a string of three random letters, e.g. "a b c", and the output is its reverse "c b a". For repeat, the input contains possibly duplicated letters, e.g. "a a b b c", and the output is the string without duplication, i.e. "a b c". This kind of letter-string logical tasks remains a challenge for current AI systems (Mitchell, 2021). For the *common-sense reasoning* task, we leverage question-answer pairs from the CommonsenseQA dataset (Talmor et al., 2019), where each pair forms an ICL demonstration. For the *mathematical arithmetic task*, we use questions and answers from the AddSub dataset (Hosseini et al., 2014).

In all tasks, the input and the output in a demonstration is separated by an arrow "→", e.g. a demonstration in the reverse task should be "a b c → c b a". Demonstrations are in turn separated by ";\n". Finally, the query consists of the input string and the arrow, e.g. "x y z →".

## 2.3 EXPERIMENTAL LLMS

We conduct experiments on various decoder-only open-source LLMs, including GPT-2 (Radford et al., 2019), GPT-Neo (Gao et al., 2020), Llama-v1 (Touvron et al., 2023a) & Llama-v2 (Touvron et al., 2023b), MPT (Team, 2023), Phi-2 (Javaheripi et al., 2023), and Qwen-2.5 (Hui et al., 2024).

## 3 PROMPT CLUSTERING

In this section, we introduce and verify the clustering property: $x_{-1}$*'s of prompts with the same first and last demonstrations tend to stay closer in the embedding space*, with first-demonstration clustering showing notably stronger effects. This means the embedding space contains clusters, each of which is associated with prompts with the same first and last demonstrations, though the clustering by first demonstration is more pronounced. We will show that the clustering property consistently occurs across different LLMs and ICL tasks.

Formally, consider a set of $k$ demonstrations $\{e_1, \ldots, e_k\}$ and a query $q$. We consider $k = 5$ in this section. Let $\mathcal{P}$ be the permutation set of $\{e_1, \ldots, e_k\}$, hence $|\mathcal{P}| = k!$. Each permutation $p \in \mathcal{P}$ and the query $q$ compose a prompt. Let $x_{-1}^p \in \mathbb{R}^d$ be the last-layer embedding of the last input token of the prompt associated with permutation $p$. We verify the clustering property of the set $\{x_{-1}^p : p \in \mathcal{P}\}$ with different approaches: low-dimensional visualization (Section 3.1), K-Means clustering and quantifying token importance (Section 3.2). Experimental results consistently confirm the occurrence of clusters. We also show that clustering appears from early layers of LLMs (Section 3.3).

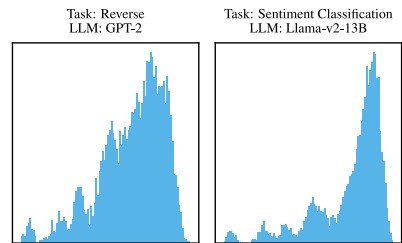

Figure 2: *Histogram of pairwise distances of points in* $\{x_{-1}^p : p \in \mathcal{P}\}$. *Both histograms contains more than one peak, indicating the existence of clusters.*

## 3.1 LOW-DIMENSIONAL VISUALIZATIONS OF CLUSTERING

Figure 1 shows the 2D UMAP (McInnes et al., 2018) projections of $x_{-1}^p$'s with various LLMs and ICL tasks (similar results for t-SNE projection are shown in Appendix E). Projections with the same colors are of permutations sharing the first demonstrations. Similarly, projections with the same shapes are of permutations sharing the last demonstrations.

We observe a consistent property across different cases: projections are clustered by colors and shapes, i.e. if $p$ and $p'$ share the first or last demonstrations, then the projections of $x_{-1}^p$ and $x_{-1}^{p'}$ tend to stay closer. This also means when $p$ and $p'$ have different first and last demonstrations, $x_{-1}^p$ and

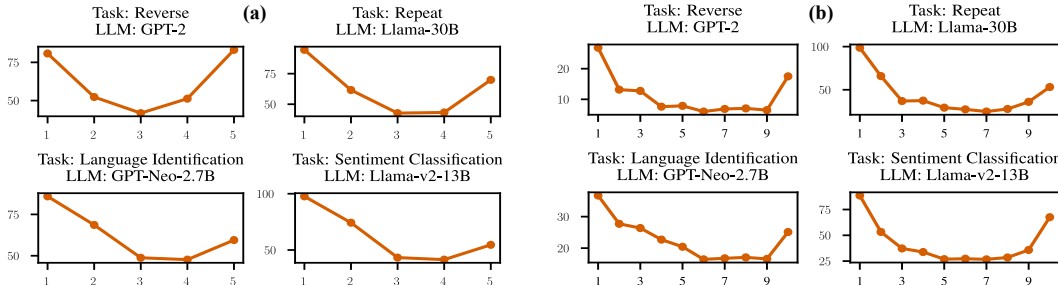

Figure 3: **(a)** *Percentage frequencies at demonstration positions.* The $x$-axis is the index of demonstration positions; $y$-axis is the percentage frequency. We observe U-shapes of the percentage frequency curves across different tasks and LLMs. This indicates first and last demonstrations of elements in a cluster tend to be the same. **(b)** *Partial derivative norms of various LLMs on different tasks.* The $x$-axis is the chunk index; $y$-axis is the chunk-averaged Frobenius norms of partial derivatives. We can observe consistent U-shapes across different LLMs and tasks, meaning that beginning tokens and ending tokens play important roles in a prompt.

$x^{p'}_{-1}$ are likely to stay in different clusters and thus far away from each other. Since $x_{-1}$ is the input for the prediction head, this indicates the next-token prediction of $p$ is likely to differ from one of $p'$. This explains why LLMs' responses vary with demonstration orders. Interestingly, our analysis reveals that while both types of clustering are present, the clustering effect tends to be stronger for shared first demonstrations compared to shared last demonstrations.

## 3.2 CLUSTERING ON HIGH DIMENSIONAL SPACES

We verify the clustering property of $x_{-1}$'s on $\mathbb{R}^d$. To do so, we employ two different methods: $K$-Means clustering, and quantifying token importance. Experimental results consistently support the clustering property, while also confirming the relatively stronger clustering effect for shared first demonstrations.

**K-Means clustering**  First, we examine the existence of clusters in $\{x^p_{-1} : p \in \mathcal{P}\}$ based on the idea in (Steinbach et al., 2004): we plot the histograms of pairwise Euclidean distances between points in $\{x^p_{-1} : p \in \mathcal{P}\}$. If the histogram has more than one peak, $\{x^p_{-1} : p \in \mathcal{P}\}$ is likely to contain clusters. Figure 2a shows the histograms of the reverse task on GPT-2 and the sentiment classification task on Llama-v2-13B. Both histograms have more than one peak, meaning clusters exist in $\{x^p_{-1} : p \in \mathcal{P}\}$.

Next, we investigate how $\{x^p_{-1} : p \in \mathcal{P}\}$ is clustered. We run $K$-Means clustering algorithm (Lloyd, 1982) with $K = k(k-1)$ centers. In each cluster, we compute the percentage frequency of the most frequent demonstration at each demonstration position. For example, with $k = 5$, each cluster is associated with a 5-dimensional array of percentage frequencies, e.g. $(80\%, 40\%, 30\%, 30\%, 55\%)$. This means at most $80\%$ of the permutations in the cluster share the same first demonstrations, at most $40\%$ of them share the same second demonstrations, and so on. We then average the percentage frequency arrays over all $K$ clusters to come up with a $k$-dimensional array representing the current prompt. Finally, we take the average of those $k$-dimensional arrays over 100 prompts with different LLMs and tasks. Figure 3a shows the results. Overall, we observe U-shape percentage frequency curves across different tasks and LLMs, with notably higher percentages for the first position. This indicates both first and last demonstrations of permutations in a cluster tend to be the same, with a stronger effect for first demonstrations, which supports our clustering property.

**Token importance**  The property of clustering by first and last demonstrations suggests beginning and ending tokens play important roles. We thus verify the clustering property by quantifying token importance. Specifically, we quantify how much $x_{-1}$ will change when each $x_i(0)$ changes: the more $x_{-1}$ is affected, the more important $x_i(0)$ is. A popular quantity to measure the input-output sensitivity is the partial derivative norm $\left\| \frac{\partial x_{-1}}{\partial x_i(0)} \right\|$ (Novak et al., 2018), where $\|.\|$ is the Frobenius norm. If the clustering property holds, the partial derivative norms of beginning and ending tokens should be the highest among all tokens.

We randomly select 50 prompts and compute the partial derivative norms with respect to their tokens. Here we omit the last input token $x_{-1}(0)$ from consideration, since perturbing it obviously leads to

Figure 4: *2D UMAP projections of $x_{-1}(t)$'s with $t \geq 0$ on reverse task and GPT-2.* Layer 0 is the input for GPT-2, while layer 48-th is the last layer of GPT-2. We observe an interesting property that $x_{-1}(t)$'s are clustered from early layers. A theoretical explanation will be given in Section 4.1.

the most dramatic changes in $x_{-1}$. As prompt lengths are different, we divide each prompt into 10 chunks: the first chunk includes the first 10% tokens, and so on, to the last chunk including the last 10% tokens. For each chunk, we compute the partial derivative norms with respect to all of its tokens, then averaging out to obtain a representative norm for that chunk. Doing this way, we always have 10 chunk partial derivative norms for a prompt, regardless of how long the prompt is.

Figure 3b shows the results. We observe consistent U-shapes across LLMs and tasks, indicating that both beginning and ending tokens play important roles in prompts, with beginning tokens showing relatively stronger effects. These results again support the existence of clustering in $\mathbb{R}^d$. Note that these results do *not* imply that middle tokens are not important (see Table 8 in Appendix F).

### 3.3 CLUSTERING AT INTERMEDIATE LAYERS

In previous sections, we have merely studied the behaviors of $x_{-1}$'s and discovered their clustering property. We wonder if the clustering property still occurs at intermediate Transformer layers. In this section, we conduct similar experiments with the 2D UMAP projections, this time for the last input token embeddings at intermediate layers (i.e. $x_{-1}(t)$'s with $t < T$). We report the results of sentiment classification task on GPT-2 in Figure 4. Interestingly, the clustering property appears from early Transformer layers, and it is maintained in the afterwards layers.

The emergence of clustering in early LLM layers suggests Transformer operations themselves may significantly contribute to clustering. In the next section, we provide theoretical and empirical evidences to support this hypothesis.

## 4 MECHANISMS BEHIND CLUSTERING PROPERTY

We are motivated by findings in Section 3 to further explore the mechanisms behind the clustering property. In this section, we provide theoretical analysis (Section 4.1) and empirical experiments (Section 4.2) to explain the observed behaviors of clustering. Our analysis suggests that while prompts with shared first or last demonstrations form clusters in the embedding space, the underlying mechanisms may differ. The first-demonstration clustering appears to be partially explained by the causal attention mask mechanism. In contrast, the clustering by last demonstrations seems to involve a more complex interaction between the causal structure and positional encoding. This difference in mechanisms aligns with our empirical observation that first-demonstration clustering tends to be more pronounced, though further investigation would be needed for a complete understanding.

### 4.1 THEORETICAL ANALYSIS

We provide theoretical evidence for the clustering property. More importantly, we would like to point out that causal attention mask is one important factor leading to clustering by beginning tokens. This, in turn, results in the property of clustering by the first demonstration.

Suppose $x_i(t)$ is on the unit sphere $\mathbb{S}^{d-1}$ for every $i = \overline{1, n}$ and $t \geq 0$. The dynamics of Transformer can be written as

$$\frac{\mathrm{d}x_i}{\mathrm{d}t} = \mathbf{P}_{x_i(t)} \left( \frac{1}{Z_{\beta,i}(t)} \sum_{j=1}^{n} e^{\langle Q(t)x_i(t), K(t)x_j(t) \rangle} V(t)x_j(t) \right), \tag{1}$$

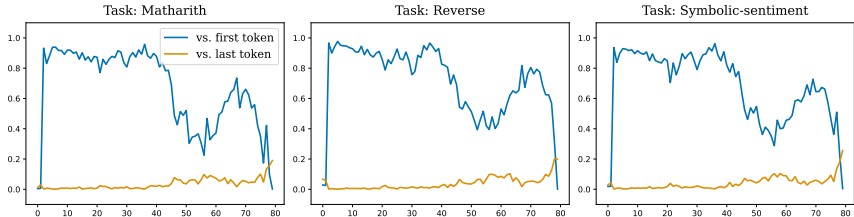

Figure 5: *Attention weights from the last token to the first token (blue) and last token (orange) with Qwen-2.5-72B across layers and different tasks.* The x-axis shows layer indices ranging 0-80; y-axis shows attention weights averaged over 100 prompts. Notably, attention to first token dominates early layers (0.8-0.9) aligning with Proposition 4.1, while attention to last token steadily increases and peaks in final layers (around 0.2). **(Left)** Mathematical arithmetic task. **(Middle)** Reverse task. **(Right)** Symbolic sentiment classification task.

for $i = \overline{1,n}$ and $t \geq 0$. Here $Q(.)$, $K(.)$, and $V(.)$ are the query, key, and value matrices, $\mathbf{P}_x y$ is the projection of $y \in \mathbb{S}^{d-1}$ onto the tangent space $\mathrm{T}_x \mathbb{S}^{d-1}$, and $Z_{\beta,i}(t) := \sum_{k=1}^{n} e^{\beta \langle Q(t)x_i(t), K(t)x_k(t) \rangle}$ is the partition function. We further assume $V(t)$ is identity for all $t \geq 0$.

Consider an initial states $\{x_i(0)\}_{i=1}^{n} \in \left(\mathbb{R}^d\right)^n$ that satisfies the following hypothesis $(H)$: there exists $w \in \mathbb{S}^{d-1}$ such that $\langle w, x_i(0) \rangle > 0$ for all $i = \overline{1,n}$. Under the hypothesis $(H)$, it has been proved that if $\{x_i(.)\}_{i=1}^{n} \in C^0 \left(\mathbb{R}_{\geq 0}; \left(\mathbb{S}^{d-1}\right)^n\right)$ is the unique solution of the corresponding Cauchy problem (1), then there exists $x^* \in \mathbb{S}^{d-1}$ and constants $C, \lambda > 0$ such that $\|x_i(t) - x^*\| \leq Ce^{-\lambda t}$ (Geshkovski et al., 2023). This means all items in the sequence $\{x_i(t)\}_{i=1}^{n}$ become identical exponentially fast w.r.t. the depth $t$ under appropriate conditions. We call $x^*$ the *convergence point* of the Cauchy problem corresponding to the initial state $\{x_i(0)\}_{i=1}^{n}$. The following proposition shows that the convergence points of two different initial states with the same first token coincide when casual attention mask is applied.

**Proposition 4.1.** Let $\{x_i(0)\}_{i=1}^{n}$ and $\{x_i'(0)\}_{i=1}^{n'}$ be different initial states satisfying $(H)$ and $x_1(0) = x_1'(0)$. Note that $n$ may be different from $n'$. Let $x^*$ and $x'^*$ are the convergence points of the Cauchy problems Eq. (1) corresponding to $\{x_i(0)\}_{i=1}^{n}$ and $\{x_i'(0)\}_{i=1}^{n'}$, respectively. If causal attention mask is applied, then $x^* = x'^*$. This means all states of the two Cauchy equations corresponding to $\{x_i(0)\}_{i=1}^{n}$ and $\{x_i'(0)\}_{i=1}^{n'}$ eventually coincide, even though they only share the same first input token.

*Proof.* The key notice here is if causal attention mask is applied, then $x_1(t) = x_1'(t)$ for all $t \geq 0$. This leads to

$$\|x^* - x'^*\| \leq \|x^* - x_1(t)\| + \|x_1'(t) - x'^*\|$$
$$\leq Ce^{-\lambda t} + C'e^{-\lambda' t}, \quad \forall t \geq 0,$$

which means $x^* = x'^*$. $\qquad \square$

Proposition 4.1 is closely related to the clustering-by-first-demonstration property. Concretely, if two prompts share the same first demonstrations, they also share the same first token (i.e. $x_1(0) = x_1'(0)$. Under appropriate circumstances as in Proposition 4.1, the convergence points of these prompts coincide, which means their associated $x_{-1}$'s also coincide if there are infinitely many Transformer layers. Moreover, with $x^* = x'^*$, we further have

$$\|x_{-1}(t) - x_{-1}'(t)\| \leq \|x_{-1}(t) - x^*\| + \|x_{-1}'(t) - x'^*\|$$
$$\leq Ce^{-\lambda t} + C'e^{-\lambda' t}, \quad \forall t \geq 0,$$

which means the last embeddings become identical exponentially fast. This explains why clustering occurs from early layers of Transformer as shown in Section 3.3.

Proposition 4.1 draws a close theoretical relationship between causal attention mask and first-demonstration clustering. We further empirically investigate this connection in the next section, together with the one between positional encoding and last-demonstration clustering.

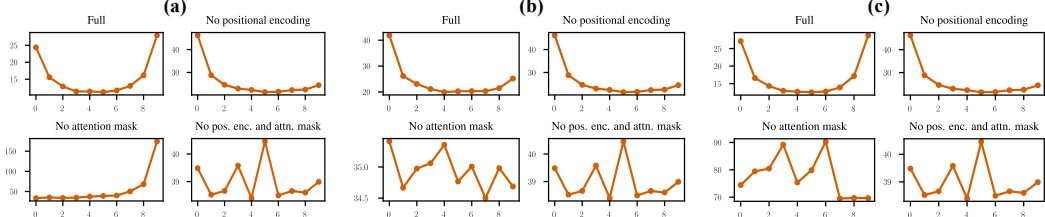

Figure 6: *Partial derivative norms w.r.t. chunks in trained-from-scratch Transformers with different types of positional encodings*. The $x$-axis is chunk indices ranging 0-9; $y$-axis is the partial derivative norms. **(a)** Sinusoidal positional encoding. **(b)** Rotary positional encoding. **(c)** Trainable positional encoding.

**Does Proposition 4.1 violate last-demonstration clustering?** While Proposition 4.1 suggests that first-token clustering is theoretically possible under ideal conditions, it does not address or contradict our empirical observation of last-demonstration clustering. Our analysis of attention weights across layers reveals a more nuanced picture of how positional information is processed in practice.

Figure 5 shows the attention weights from the first token (blue) and last token (orange) to the last token across different layers, measured on the Qwen-2.5-72B model (Hui et al., 2024). The results are averaged over 100 different prompts of varying lengths across multiple tasks including mathematical arithmetic, reverse and symbolic sentiment classification. This comprehensive sampling ensures our findings are robust across different task types and input lengths. The attention pattern exhibits three distinct phases:

- Initial Phase (layers 1-40): The first token dominates attention with consistently high weights (0.8-0.9), showing remarkable stability in early layers. This strong initial convergence precisely aligns with our theoretical prediction in Proposition 4.1 and reflects the attention sink phenomenon (Xiao et al., 2024).
- Transition Phase (layers 40-60): We observe a sharp decline in first-token attention, dropping from 0.8-0.9 to 0.2-0.4. During this phase, attention begins to redistribute across tokens, marking a shift from the theoretical behavior to practical requirements.
- Final Phase (layers 60-80): The first-token attention exhibits oscillating patterns (between 0.4-0.7), while attention to the last token steadily increases and reaches its peak (0.1-0.2) in the final layers. This consistent pattern of increasing last-token attention across tasks, despite variations in first-token attention, suggests a fundamental architectural behavior aligned with the next-token prediction objective.

This layerwise progression offers insights into how our theoretical and empirical findings coexist: while the theoretical tendency toward first-token clustering manifests in early layers (with attention weights of 0.8-0.9), the practical requirements of causal language modeling lead to attention redistribution in later layers, where we observe increased attention to the last token (reaching 0.1-0.2). This pattern suggests a dynamic balance between the model's architectural bias toward first-token clustering and its need to capture sequence-final information for next-token prediction. This observation aligns with prior findings that causal attention mechanisms can infer positional information even without explicit positional encoding (Kazemnejad et al., 2024) - a property that emerges naturally from the causal structure. When combined with positional encodings, this evolving attention pattern appears to contribute to the observed dual clustering behavior - both first-demonstration and last-demonstration clustering - though our experiments in Section 4.2 suggest this interaction is complex and dependent on the specific type of positional encoding used.

## 4.2 EMPIRICAL ANALYSIS

While our theoretical analysis in Section 4.1 suggests how causal attention mask could enable first-demonstration clustering under ideal conditions, we need to examine how these mechanisms manifest in practice without theoretical restrictions. We investigate both first-demonstration and last-demonstration clustering through empirical experiments, following a similar methodology to Section 3.2. Our experiments suggest that different architectural components contribute distinctly to clustering behavior: the causal attention mask appears particularly important for first-demonstration clustering, while both causal attention mask and positional encoding seem to play roles in last-demonstration clustering.

|            | Exhaustive   | Cluster | *Relative decrease* |
|------------|--------------|---------|---------------------|
| GPT-Neo-2.7B | **97.4** (1.8)  | 94.0    | 3.5                 |
| Llama-v2-7B  | **98.5** (4.5)  | 95.6    | 2.9                 |
| MPT-7B       | **98.7** (11.9) | 97.8    | 0.9                 |

Table 1: *Accuracies (%) of Exhaustive and Cluster-based Search with the ideal selecting criterion with different LLMs on reverse task.* The first two columns show accuracies, while the third one shows the relative decreased percentage of Cluster-based Search compared to Exhaustive Search. Exhaustive Search achieves the best accuracies on all LLMs, but the performances of Cluster-based Search are comparable. Numbers in parentheses are accuracies with the dumb selection criterion.

To investigate the role of causal attention mask and positional encoding, we train different Transformers with or without those components. Specifically, we train from scratch four Transformers: one with full components, one without positional encoding, one without causal attention mask, and one without both. Other Transformer components are always included. Each Transformer has 12 self-attention layers, each with 12 attention heads; the token embedding size is 768, and the hidden size in MLP layers is 2048. We train the Transformers with different types of positional encodings, namely the sinusoidal, rotary, and trainable positional encoding, on WikiText2 dataset with SGD optimizer with learning rate $5 \cdot 10^{-1}$ in 100 epochs. For fair comparisons, the training task for all Transformers is next-token prediction.

We prepare 100 randomized prompts and compute the partial derivative norms similarly to Section 3.2. To ensure the prompts are differently distributed to training ones, we build each prompt as a sequence of 50 to 100 random words, resulting in meaningless sentences. For each prompt, we compute its chunk partial derivative norms, then average over 100 prompts. Figure 6 shows interesting results. It reveals a robust correlation between first-demonstration clustering and the utilization of the causal attention mask. Specifically, the importance of beginning tokens is markedly elevated when, and only when, the causal attention mask is applied, which aligns with the findings presented in Proposition 4.1. On the other hand, the case for last-demonstration is more complex. While the importance of ending tokens remains distinctively high when sinusoidal positional encoding is employed in the absence of a causal attention mask, this phenomenon is not observed for rotary and trainable positional encoding. This suggests that the importance of ending tokens is influenced by the interplay between the causal structure and the choice of positional encoding method.

## 5 Accelerating Selection and Ordering of In-Context Demonstrations in Self-Adaptive ICL Settings

Based on the clustering property, we propose an efficient approach to improve self-adaptive ICL methods (Wu et al., 2022). In particular, self-adaptive methods aim to optimize the selection and ordering of demonstrations based on model's own predictions, without relying on external knowledge or supervision. However, this process typically suffers from factorial complexity. For example, when exhaustively selecting and ordering $k$ out of $k_{\text{total}}$ demonstrations (referred to as *Exhaustive Search*), there are $A_{k_{\text{total}}}^{k} := \frac{k_{\text{total}}!}{(k_{\text{total}}-k)!}$ possibilities. In contrast, the first-demonstration clustering property suggests that prompts sharing the first demonstration are likely to have the same next-token prediction. Consequently, our approach, called *Cluster-based Search*, only requires selecting the first demonstration, while the rest can be randomly selected, resulting in merely $k_{\text{total}}$ possibilities.

Formally, consider an LLM, a pool of $k_{\text{total}}$ demonstrations $E$ and a query $q$. Any algorithm for demonstration selection and ordering will construct a set of possible prompts, $\mathcal{P}(E, q)$, and uses a pre-defined criterion $\mathcal{C}$ to select among this set a potentially correct prompt, i.e. one with which the LLM can answer the query correctly. In this section, we consider two types of criterion $\mathcal{C}$: the ideal criterion and the practical entropy-based one. In both cases, Cluster-based Search achieves comparable performances with Exhaustive Search while being significantly faster.

We consider $k_{\text{total}} = 6$ and $k = 4$ for as it is feasible to run Exhaustive Search. We also report results when $k_{\text{total}} = 16$ and $k = 4$ in Appendix D to show the scalability of Cluster-based Search. Furthermore, in this section, we focus on non-instructed prompt to better align with previous

|  |  | Classification | | | Reasoning | | | Avg |
|---|---|---|---|---|---|---|---|---|
|  |  | SymSen | SymLan | Rev. | Rep. | ComSen | Math | |
| GPT-Neo-2.7B | Random | $51.3_{4.7}$ | $65.2_{4.0}$ | $56.6_{5.1}$ | $49.5_{4.1}$ | $17.3_{3.1}$ | $1.0_{0.8}$ | 40.2 |
|  | Exhaustive | 51.5 | 77.0 | 66.5 | 49.0 | 18.7 | 0.5 | 43.9 |
|  | Cluster | $51.5_{5.5}$ | $74.6_{3.3}$ | $76.3_{5.1}$ | $55.3_{5.6}$ | $16.6_{3.5}$ | $1.0_{0.8}$ | **45.9** |
| Phi-2 (2.7B) | Random | $23.1_{3.7}$ | $59.3_{4.8}$ | $77.9_{4.9}$ | $53.0_{3.4}$ | $61.2_{4.0}$ | $75.1_{4.0}$ | 58.3 |
|  | Exhaustive | 31.5 | 76.0 | 80.5 | 71.0 | 63.5 | 77.0 | **66.6** |
|  | Cluster | $30.8_{3.7}$ | $72.3_{5.1}$ | $77.3_{3.5}$ | $65.0_{4.6}$ | $64.2_{3.9}$ | $77.9_{4.2}$ | 64.6 |
| Qwen-2.5-14B | Random | $71.7_{3.0}$ | $76.3_{2.8}$ | $99.1_{0.8}$ | $93.3_{2.4}$ | $84.3_{3.4}$ | $85.1_{3.2}$ | 85.0 |
|  | Exhaustive | 78.0 | 86.0 | 100.0 | 96.0 | 84.2 | 85.5 | 88.3 |
|  | Cluster | $82.3_{2.9}$ | $87.1_{3.3}$ | $99.3_{0.9}$ | $97.7_{1.6}$ | $84.3_{2.5}$ | $87.1_{3.0}$ | **89.6** |

Table 2: *Accuracies (%) of Random Selection, Exhaustive Search, and Cluster-based Search with entropy-based selecting criterion on different LLMs and ICL tasks.* The subscript numbers indicate the standard deviation over 10 runs. Due to computational constraints, we do not report standard deviations for Exhaustive Search. **Bold**: best; Underline: second best. Searching is obviously more effective than random selection. Moreover, the performances of Cluster-based Search are comparable to Exhaustive Search.

discussions on the positions of demonstrations. Additional results regarding the instructed-prompt case are given in Table 7 in Appendix F.

## 5.1 IDEAL SELECTION CRITERION

In this section, we consider $\mathcal{C}$ to be the ideal criterion, which always selects the correct prompt in $\mathcal{P}(E, q)$ if exists. This criterion is alternatively called the oracle (Lu et al., 2022). The accuracy achieved by this ideal criterion sets an upper bound for any other selecting criterion. We also consider the dumb criterion, which always select the wrong prompt if exists. In the following, we show the accuracy upper bounds of Cluster-based Search are comparable to Exhaustive Search while significantly reducing the search time.

Accuracies of different LLMs on reverse task are reported in Table 1. Here we take the average accuracy over $1,000$ tuples of $(E, q)$. See Appendix C for results of other tasks. While Exhaustive Search achieves the best accuracies on all LLMs, it is worth noticing that the performances of Cluster-based Search are comparable. Concretely, performances of Cluster-based Search only decrease relatively by $2.4\%$ on average compared to Exhaustive Search, while search time of Cluster-based Search decreases by $91.7\%$ (Figure 7). Note that with larger $k_{\text{total}}$, the time saving is almost $100\%$. Additionally, the perfor-

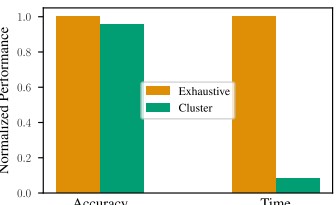

Figure 7: *Normalized accuracies and search times of Exhaustive Search and Cluster-based Search with the ideal selecting criterion.*

mance gaps between the ideal and the dumb criterion (number in parentheses in Table 1) are huge for all LLMs, further highlighting the efficiency of Cluster-based Search.

## 5.2 ENTROPY-BASED SELECTION CRITERION

We consider a more practical selecting criterion $\mathcal{C}$, i.e. the popular entropy-based criterion (Lu et al., 2022), to show the effectiveness of Cluster-based Search in practice. To be concrete, denote $\ell_{-1}$ to be the logits of the first prediction step. The size of $\ell_{-1}$ is equal to the length of the token dictionary, and it is the output of the prediction head whose input is $x_{-1}$. We define the *confidence score* of $\ell_{-1}$ to be $c(\ell_{-1}) \coloneqq -\text{entropy}(\text{softmax}(\ell_{-1}))$. A permutation $p$ of demonstrations is more confident than another permutation $p'$ if its associated logits $\ell_{-1}$ is more confident than $\ell'_{-1}$, i.e. $c(\ell_{-1}) > c(\ell'_{-1})$. Among all prompt candidates, we will select the most confident one. See Algorithm 1 in Appendix B for the pseudocode of this criterion.

We run the Exhaustive Search and Cluster-based Search with Algorithm 1 on various ICL tasks introduced in Section 2.2. Each task includes $1,000$ tuples $(E, q)$. Average accuracies of different LLMs are reported in Table 2. On average, the performances of Random Selection are the worst, with large gaps compared to Exhaustive Search, showing the effectiveness of selecting and ordering.

| | | Classification | | | Reasoning | | | Avg |
|---|---|---|---|---|---|---|---|---|
| | | SymSen | SymLan | Rev. | Rep. | ComSen | Math | |
| GPT-Neo-2.7B | First Cluster | 51.5 | 74.6 | 76.3 | 55.3 | 16.6 | 1.0 | **45.9** |
| | First-Last Cluster | 55.5 | 75.7 | 63.7 | 53.2 | 17.8 | 1.0 | 44.5 |
| Phi-2 (2.7B) | First Cluster | 30.8 | 72.3 | 77.3 | 65.0 | 64.2 | 77.9 | 64.6 |
| | First-Last Cluster | 28.1 | 72.2 | 78.9 | 69.3 | 62.7 | 78.5 | **64.9** |
| Qwen-2.5-14B | First Cluster | 82.3 | 87.1 | 99.3 | 97.7 | 84.3 | 87.1 | **89.6** |
| | First-Last Cluster | 79.3 | 83.7 | 99.2 | 97.0 | 84.0 | 86.2 | 88.2 |

Table 3: Comparison between First-only and First-Last clustering search strategies across different models and tasks, showing accuracy (%) on classification and reasoning tasks.

However, it is interesting to note that performances of Cluster-based Search are comparable to Exhaustive Search, with a slight absolute performance gap of $0.3\%$ on average. This result once again highlights the efficiency of Cluster-based Search.

### 5.2.1 ABLATION STUDIES

We further investigate whether selecting only the first demonstration, rather than both first and last demonstrations, could maintain performance while improving efficiency. Table 3 presents the results of this investigation.

The results demonstrate that selecting only the first demonstration for clustering-based search achieves comparable or sometimes better accuracy compared to selecting both first and last demonstrations. This is particularly evident in GPT-Neo-2.7B where first-only clustering achieves an average accuracy of 45.9% compared to 44.5% for first-last clustering, and in Qwen-2.5-14B where it achieves 89.6% versus 88.2%. Most notably, this simpler approach reduces the computational complexity significantly - from $O(k_{\text{total}}(k_{\text{total}} - 1))$ when selecting both first and last demonstrations to just $O(k_{\text{total}})$ when selecting only the first demonstration. This order of magnitude improvement in efficiency, combined with the maintained or improved accuracy, makes first-only clustering a clearly superior choice for demonstration selection.

## 6 RELATED WORK

Recent research has unveiled the sensitivity to the order of in-context demonstrations in large language models (LLMs). It has demonstrated that simply reshuffling the examples in a sentiment analysis prompt can dramatically impact accuracy, ranging from near-random guessing to state-of-the-art performance (Zhao et al., 2021). Subsequent studies have confirmed this "order sensitivity" is widespread, affecting models of all sizes and independent of specific examples (Liu et al., 2021). Further work noted a negative correlation between sensitivity and accuracy, suggesting highly sensitive outputs are less reliable (Chen et al., 2022). Another study finds that language models' performance can significantly decline when relevant information appears in the middle of the context, suggesting they may struggle to effectively utilize long contexts (Liu et al., 2023). To the best of our knowledge, our paper is the first to investigate the prompt embeddings to explain the order sensitivity of LLMs in ICL. Furthermore, we also provide theoretical and empirical evidences to support our findings.

Oversmoothing is a problem where information is lost and representations become identical. As explored in Section 3.2, this issue is closely related to the clustering property of LLMs. Previous work shows Transformer is prone to oversmoothing during training, especially with deep architectures, and this ultimately lowers model performance (Chen et al., 2022). Further research finds the underlying cause of oversmoothing in Transformer, and hypothesizes that layer normalization plays a crucial role in oversmoothing (Shi et al., 2022) . Recently, a phenomenon called attention sink has been discovered, which states that a large portion of the attention weights stays at the first token, starting from early layers of the Transformer (Xiao et al., 2024) . This may partly explain the oversmoothing problem in Transformer.

The feature importance in neural networks can be assessed by the Jacobian matrix, which maps minor input changes to output shifts. This sensitivity measure has motivated research on model robustness and knowledge transfer. Recent work reveal that even the simple optimization algorithm SGD subtly

regularizes the model's sensitivity (Lee et al., 2023) , while other studies manipulate the Jacobian to boost noise and attack resistance (Hoffman et al., 2019) . Beyond robustness, it has been highlighted how Jacobian regularization supports more realistic dynamics in the system (Finlay et al., 2020).

The serial-position effect, a fascinating psychological phenomenon where first and last items in a sequence are recalled best, has been a fruitful area of study. Pioneering work highlights how position influences recall, suggesting more rehearsal for beginning and ending items (Murdock and Bennet, 1962). Later research identifies two driving forces: the primacy effect, boosting recall for initial items, and the recency effect, favoring recent ones (Glanzer and Cunitz, 1966). We give a brief discussion on the connection between the serial-position effect and our clustering property in Appendix A, and we believe that this research direction is interesting to explore further.

## 7 CONCLUSION AND LIMITATIONS

We study the prompt embedding space to understand the order sensitivity of ICL in decoder-only LLMs. Our analysis reveals that prompts sharing first or last demonstrations tend to form clusters in the embedding space, with first-demonstration clustering showing notably stronger effects. Our theoretical analysis suggests this asymmetry may be partially explained by the causal structure, as first-token information is reused $O(n^2)$ times in a self-attention layer. The last-demonstration clustering appears to emerge through a more complex interaction between the causal structure and positional encoding, though its precise mechanisms warrant further investigation. Based on these insights, we introduce Cluster-based Search for demonstration selection and ordering in self-adaptive ICL methods, achieving comparable performances to exhaustive search while reducing computational complexity from factorial to quadratic.

Our work contributes to understanding the operational mechanisms of LLMs, an area that would benefit from increased research attention. The clustering property we identify may represent one of several important characteristics that could enhance our comprehension of these models. Such insights can lead to practical improvements in LLM applications, as demonstrated by our Cluster-based Search method.

Several limitations of this study should be acknowledged. First, while Cluster-based Search shows promise in self-adaptive settings, its effectiveness in other contexts requires further evaluation. Second, our theoretical framework primarily addresses first-demonstration clustering, while the mechanisms behind last-demonstration clustering remain incompletely understood. Third, verifying the clustering property poses challenges in black-box models where internal representations are not accessible. Future research should focus on developing techniques to better understand these behaviors and extend our findings to broader contexts.

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

# APPENDIX

## A    RELATION OF CLUSTERING PROPERTY AND SERIAL-POSITION EFFECT

Analyses in Section 4.1 and 4.2 allow us to link the clustering property with the well-known serial-position effect (Murdock and Bennet, 1962), which has two components: primacy effect and recency effect. The *primacy effect* means that the first item in a sequence is remembered better than middle ones. Previous research attributes this to the long-term memory: the first item gets more rehearsal when the person tries to recall the sequence, and thus it is more likely to be stored in the long-term memory (Rundus, 1971). This is similar to causal attention mask, where all tokens attend to the first one for updating. Moreover, it has been shown that that the primacy effect weakens with longer sequences (Murdock and Bennet, 1962). We see the same pattern in LLMs: the ratio between the importance of beginning tokens and ending tokens shrinks as the prompts get longer (see Figure 8). On the other hand, the *recency effect* means that the last item in a sequence is remembered better than middle ones. This is due to the short-term memory, which keeps a few recent items (Craik et al., 1970). Recent work suggests that positional encoding adds this locality to language models (Chen et al., 2023). Following Secion 4.2, this implies that the last-demonstration clustering property is very similar to the recency effect.

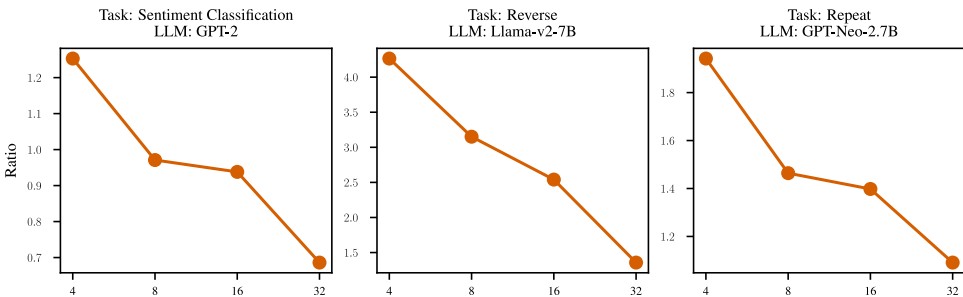

Figure 8: *Ratio of the first and last token chunk importance*. The $x$-axis is the number of demonstrations. The ratios consistently decrease with different LLMs and ICL tasks.

## B    ENTROPY-BASED SELECTION CRITERION PSEUDO-CODE

Here we give the pseudo-code for the entropy-based selection criterion presented in Section 5.2. The algorithm computes the confidence score for each permutation of demonstrations based on the entropy of the softmax output of the first prediction step's logits. It then selects the permutation with the highest confidence score, i.e., the lowest entropy, as the most confident prompt candidate. This approach allows for the identification of the optimal demonstration order that maximizes the model's confidence in its predictions.

---

**Algorithm 1** Entropy-Based Selecting Criterion

---

**Input:** set of prompt candidates $\mathcal{P}$
**set** cBest $= -$inf
**set** pBest $=$ None
**for** $p$ **in** $\mathcal{P}$ **do**
    compute logits $\ell_{-1}$
    compute confidence score $c\left(\ell_{-1}\right)$
    **if** $c\left(\ell_{-1}\right) >$ cBest **then**
      cBest $= c\left(\ell_{-1}\right)$
      pBest $= p$
    **end if**
**end for**
**Output:** pBest

---

| | | Classification | | | | Reasoning | | Avg |
|---|---|---|---|---|---|---|---|---|
| | | Sent. | Lang. | Sym.Sen. | Sym.Lan. | Rev. | Rep. | |
| GPT-Neo-2.7B | Exhaustive | 99.9 | 100.0 | 52.0 | 100.0 | 97.4 | 94.6 | 90.7 |
| | Cluster | 99.6 | 99.8 | 51,9 | 100.0 | 94.0 | 90.3 | 89.3 |
| Llama-v2-7B | Exhaustive | 99.7 | 100.0 | 100.0 | 100.0 | 98.5 | 92.6 | 98.5 |
| | Cluster | 99.4 | 99.9 | 100.0 | 99.8 | 95.6 | 89.0 | 97.3 |
| MPT-7B | Exhaustive | 99.7 | 99.9 | 52.0 | 100.0 | 98.7 | 97.8 | 91.4 |
| | Cluster | 99.2 | 99.9 | 52.0 | 99.9 | 97.8 | 96.0 | 90.8 |

Table 4: *Accuracies (%) of Exhaustive and Cluster Search with the ideal selecting criterion with different LLMs on various task.* The case $k_{\text{total}} = 6$ and $k = 4$.

| | | Classification | | | Reasoning | | | Avg |
|---|---|---|---|---|---|---|---|---|
| | | SymSen | SymLan | Rev. | Rep. | ComSen | Math | |
| GPT-Neo-2.7B | Random | 51.0 | 54.0 | 58.0 | 47.0 | 13.0 | 2.0 | 37.5 |
| | Cluster | 55.5 | 80.0 | 82.0 | 56.5 | 17.0 | 1.5 | **48.8** |
| Phi-2 (2.7B) | Random | 22.5 | 53.0 | 77.0 | 53.5 | 57.5 | 80.0 | 57.3 |
| | Cluster | 36.0 | 82.0 | 78.5 | 69.5 | 62.5 | 76.5 | **67.5** |
| Qwen-2.5-14B | Random | 65.0 | 74.0 | 98.0 | 98.5 | 86.5 | 88.0 | 85.0 |
| | Cluster | 88.0 | 96.0 | 100.0 | 99.5 | 85.5 | 86.0 | **92.5** |

Table 5: *Accuracies (%) of Random Selection, Exhaustive Search, and Cluster Search with entropy-based selecting criterion on different LLMs and ICL tasks.* The case $k_{total} = 16$ and $k = 4$.

## C  SEARCH WITH IDEAL SELECTION CRITERION

We report in Table 4 results of similar experiments as in Section 5.1 on different LLMs and ICL tasks. By comparing the performance of Cluster-based Search with Exhaustive Search, we can assess the effectiveness of our proposed method in finding optimal demonstration orders. Overall, the performances of Cluster-based Search are comparable to Exhaustive Search, while significantly reducing the search time complexity.

## D  CLUSTER SEARCH WITH $k_{\text{TOTAL}} = 16$ AND $k = 4$

We report in Table 5 results of similar experiments as in Section 5.2 with $k_{\text{total}} = 16$ and $k = 4$. The table presents the average accuracies of different LLMs on various ICL tasks using the entropy-based selection criterion. By comparing the performances of Cluster-based Search and Random Selection, we can assess the effectiveness and scalability of our proposed method when the total number of demonstrations is increased to 16. Moreover, comparing to Table 2, we observe that the performance of Cluster-based Search is better when the pool is larger. These results provide further evidence for the efficiency of Cluster-based Search in selecting and ordering demonstrations for in-context learning, even when dealing with a larger pool of available demonstrations.

We further conduct similar experiments with $k_{\text{total}} = 16$ and $k = 10$ to show the scalability of our proposed Cluster Search. In this case, the Exhaustive Search is clearly infeasible. Experimental results are reported in Table 6. On average, the Cluster Search method is better than Random Search by $1.5\%$ to $5.9\%$.

## E  LOW-DIMENSIONAL PROJECTIONS WITH T-SNE

We re-build Figure 1 with UMAP projection replaced by t-SNE projection. The t-SNE projections, shown in Figure 9, exhibit similar clustering patterns as observed in the UMAP projections. Permutations sharing the first demonstrations (indicated by colors) or the last demonstrations (indicated by shapes) tend to form distinct clusters in the low-dimensional space. This consistency across different dimensionality reduction techniques further supports our clustering property.

|  |  | Classification | | | Reasoning | | | Avg |
|---|---|---|---|---|---|---|---|---|
|  |  | SymSen | SymLan | Rev. | Rep. | ComSen | Math |  |
| GPT-Neo-2.7B | Random | 54.1 | 75.1 | 78.9 | 62.1 | 20.6 | 2.7 | 48.9 |
|  | Cluster | 52.6 | 82.1 | 85.8 | 62.0 | 19.1 | 2.9 | **50.8** |
| Phi-2 (2.7B) | Random | 28.3 | 71.2 | 77.4 | 62.7 | 67.0 | 79.4 | 64.3 |
|  | Cluster | 38.3 | 84.8 | 80.3 | 68.1 | 67.6 | 82.0 | **70.2** |
| Qwen-2.5-14B | Random | 84.6 | 90.4 | 99.8 | 99.3 | 85.8 | 87.8 | 91.3 |
|  | Cluster | 88.5 | 95.9 | 99.6 | 99.2 | 85.5 | 88.0 | **92.8** |

Table 6: *Accuracies (%) of Random Selection, Exhaustive Search, and Cluster Search with entropy-based selecting criterion on different LLMs and ICL tasks. The case $k_{total} = 16$ and $k = 10$.*

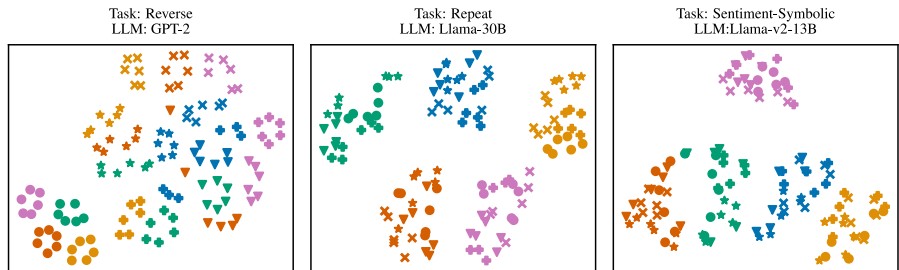

Figure 9: *t-SNE 2D projections of $x_{-1}$'s with different tasks and LLMs.* Points with the same color and shape tend to form clusters.

## F  OTHER PROBLEMS REGARDING PROMPT DESIGN

**Prompt with instruction**    In this paper, we only focus on prompts without instructions to highlight the clustering property with respect to the orders of in-context examples. However, when an instruction is included, i.e., when the first token of the prompt is a part of the instruction, our theoretical result in Proposition 4.1 still holds. Furthermore, Figure 3 also gives us evidences that the first and last demonstrations are still more important than the other ones even when the prompt starts with an instruction rather than a demonstration. We provide additional empirical evidence in Table 7 for the case of reverse task with instructed prompt. Overall, the performances of our cluster-based method are comparable to the costly exhaustive method.

|  | GPT-2 | GPT-Neo-2.7B | Llama-v2-13B |
|---|---|---|---|
| Random | 30.3 | 67.2 | 94.3 |
| Exhaustive | **37.0** | **83.8** | 94.8 |
| Cluster | 35.3 | 82.7 | **95.1** |

Table 7: *Performances of diferent methods with different LLMs in the reverse task with instructed prompt. Here we consider $k_{total} = 6$ and $k = 4$. The cluster-based method is still comparable to the exhautive method.*

**About the importance of middle in-context demonstrations**    The clustering property does not imply that the middle in-context demonstrations are not important. Indeed, it only shows that first and last demonstrations are more important than the others. However, the middle demonstrations still play a role in shaping the model's understanding of the task and the context. They provide additional examples and variety, which can help the model generalize better. Empirical evidence is given in Table 8, in which we run different LLMs on the reverse task with 4 and 8 demonstrations, respectively. The performances of the case of 8 demonstrations are consistently better, showing the importance of middle demonstrations.

|  | GPT-2 | Llama-v1-7B | Llama-v2-13B | Llama-v1-30B |
|---|---|---|---|---|
| 4 demonstrations | 32.0 | 19.7 | 95.1 | 80.4 |
| 8 demonstrations | **44.4** | **27.5** | **98.0** | **86.7** |

Table 8: *Performances of diferent LLMs on reverse task with different number of demonstrations.*

