# OpenReview forum: "Rapid Selection and Ordering of In-Context Demonstrations via Prompt Embedding Clustering"
_ICLR.cc/2025/Conference — ICLR 2025 Poster_

### Official Review · Reviewer_NLJn · 2024-10-25

**Soundness:** 2
**Presentation:** 3
**Contribution:** 2
**Rating:** 6
**Confidence:** 3

**Summary:**

This paper studied the ordering effects of demonstrations in in-context learning (ICL) and claimed that the first and last demonstrations are the most crucial ones for effective demonstrations using both empirical and theoretical analyses. Based on this observation, this paper proposed a cluster-based search method to find out effective demonstration orders (considering only the first and last demonstrations instead of all demonstrations), which will not suffer from the efficiency issue in Exhaustive Search. The experiments showed that the proposed method achieve small drop in accuracy but significant improvement in efficiency.

**Strengths:**

* The proposed idea of cluster-based search is simple yet effective for ICL.
* The performance of the proposed method, especially efficiency improvement, is very promising.

**Weaknesses:**

* Some claims are not well supported by the empirical analyses. The cluster structure of GPT-2 model in Figure 1 seems unclear, compared to the other two LLMs. Figure 3 (a) shows that the clusters also share the same second demonstrations with high percentage, and for the two bottom figures, the percentage of sharing the same second demonstrations is even higher than the percentage of sharing the same last demonstrations. These observations may be conflict with the main claim of this work. Also, the analyses about the last demonstration seem to be less convincing, e.g., lines 340-346.
* The theoretical analyses are counter intuitive. According to Prop. 4.1, the embedding of the transformer layers will eventially the same if two promopts share the same first input token. I cannot understand this claim in the proof also, in which the authors mentioned that "if causal attention mask is applied, then x_1(t) = x'_1(t) for all t >= 0." I am not sure why this assumption holds. Intuitively, if this proposition holds, I may infer that only the first demonstration will affect the performance and the last demonstration will not matter too much, which is different from the authors' claim.
* More comprehensive experiments are required. In Table 1, the case of Random demostrations is not included. It would be useful to also compare with Random ordering as in Table 2. Also, they authors used k=4 in the experiments, it might be also important to evaluate larger k values, e.g., 10 or 20. The main claim of this paper is that the demonstrations in the middle are not very important to the performance of ICL, but using only a few demonstrations in the middle (as in the experiments) may not be as convincing as using many demonstrations in the middle.

**Questions:**

Please refer to my concerns in the weakness part.

---

> ### Author Response · Authors · 2024-11-28
> **Response to Reviewer NLJn**
>
> Dear Reviewer,
>
> Thank you for your detailed and constructive feedback. We have carefully addressed each of your concerns as follows:
>
> **Empirical Analysis and Main Claims**
>
> Based on your observations, we have made significant revisions to better align our claims with the empirical evidence:
>
> 1. **Refined Main Claim**: While we maintain that both first and last demonstration clustering exist, we now explicitly emphasize that first-demonstration clustering tends to be stronger in practice. This refinement better reflects the empirical evidence across different visualizations and analyses.
>
> 2. **Improved Methodological Alignment**: Following this refined understanding, we have updated our Cluster-based Search to focus only on first-demonstration selection, rather than both first and last as in the previous version. Our ablation studies (Table 3) show this simpler approach achieves comparable or sometimes better accuracy while significantly reducing computational complexity from O(k_total(k_total-1)) to O(k_total).
>
> 3. **Evidence for Last-Demonstration Clustering**: While we acknowledge that last-demonstration clustering may appear less pronounced in some visualizations, multiple lines of evidence still support its existence:
>    - Figure 3a shows elevated percentage frequencies for last demonstrations compared to middle positions
>    - Figure 3b demonstrates higher partial derivative norms (token importance) for the last chunk versus middle chunks
>    - We've added new evidence in Figure 5 showing attention weights to the last token steadily increase across layers and peak in final layers (reaching 0.1-0.2), suggesting prompts sharing last tokens are likely to produce similar next-token predictions
>
> This more nuanced characterization better captures the asymmetric nature of demonstration importance while maintaining empirical rigor. The effectiveness of our simplified first-demonstration-only Cluster-based Search provides additional validation for this refined understanding.
>
> **Theoretical Analysis**
>
> Your questions about Proposition 4.1 highlight important technical aspects that we will clarify:
>
> The statement "if causal attention mask is applied, then x_1(t) = x'_1(t) for all t ≥ 0" follows directly from the causal attention mask mechanism. Because of causality, the first token's embeddings cannot be influenced by subsequent tokens - they can only attend to themselves. Therefore, if two sequences share the same first token (x_1(0) = x'_1(0)), their first-token embeddings will remain identical through all layers, as each layer's computation for the first token depends solely on the previous layer's first-token embedding.
>
> Your intuition about the proposition's implications for model behavior is insightful. Indeed, if the theoretical conditions held perfectly in practice, it might suggest overwhelming dominance of the first demonstration. However, we've discovered a more nuanced reality that we now better explain in Section 4.1.
>
> Specifically, we've added an analysis of attention weight patterns across layers using the Qwen-2.5-72B model that reveals three distinct phases:
>
> 1. **Initial Phase (layers 1-40)**: First-token attention dominates (0.8-0.9), aligning with Proposition 4.1 and the attention sink phenomenon
> 2. **Transition Phase (layers 40-60)**: First-token attention sharply declines from 0.8-0.9 to 0.2-0.4 as attention begins redistributing
> 3. **Final Phase (layers 60-80)**: First-token attention oscillates between 0.4-0.7, while last-token attention steadily increases to peak at 0.1-0.2
>
> This progression helps reconcile our theoretical and empirical findings: while the theoretical tendency toward first-token clustering manifests in early layers, practical requirements of causal language modeling lead to attention redistribution in later layers. When combined with positional encodings, this creates the dual clustering behavior we observe.
>
> Our refined understanding aligns with recent work showing that causal transformers can infer positional information even without explicit positional encoding, while adding positional encoding enhances this capability. This helps explain why we observe both types of clustering, though first-demonstration clustering tends to be stronger in practice. Please refer to Section 4.1 in the revision for more detail.
>
> (1/2)

---

> ### Author Response · Authors · 2024-11-28
>
> **Additional Experiments**
>
> Following your suggestion for more comprehensive experiments with larger k values, we have conducted new experiments with k_total=16 and k=10, reported in Table 6 of Appendix D. These results importantly demonstrate:
>
> 1. Cluster-based Search continues to outperform Random Selection even with a larger number of demonstrations. Specifically, the accuracy improvements over Random Selection range from 1.5% to 5.9% across different models and tasks.
>
> 2. The comparison with Exhaustive Search becomes especially relevant here - with k=10 and k_total=16, Exhaustive Search is computationally infeasible due to factorial complexity, highlighting the practical value of our approach.
>
> 3. The effectiveness of Cluster-based Search with larger k values (where more middle demonstrations are present) provides additional support for our method's robustness, while acknowledging that middle demonstrations still contribute to model performance.
>
> These new results complement our existing experiments with k=4 and help establish that our method scales effectively to scenarios with more demonstrations. Thank you for helping us make our evaluation more comprehensive.
>
> The effectiveness of our simplified approach, combined with our more nuanced theoretical understanding and comprehensive experimental validation, provides a stronger foundation for our work. Thank you for helping us improve the clarity, completeness, and rigor of our presentation. We hope that the reviewer can increase your score accordingly.
>
> (2/2)

---

> > ### Comment · Reviewer_NLJn · 2024-11-28
> > **Response to the author's rebuttal**
> >
> > I appreciate the detailed reponses from the authors, in which the new results and analyses have addressed my major concerns. Therefore, I will increase my rating to 6 and vote for acceptance.

---

### Official Review · Reviewer_nbs8 · 2024-10-31

**Soundness:** 3
**Presentation:** 2
**Contribution:** 2
**Rating:** 6
**Confidence:** 3

**Summary:**

The paper explores the issue of demonstration order sensitivity in large language models (LLMs) during in-context learning (ICL) and uncovers a clustering phenomenon in the embedding space, where prompts with the same first and last demonstrations tend to cluster together. Through theoretical analysis and empirical evidence, the paper identifies that this clustering effect stems from the interaction of causal attention masks and positional encoding. Moreover, they propose a "Cluster-based Search" method that significantly reduces the computational complexity of selecting and ordering demonstrations while maintaining high model performance.

**Strengths:**

1. Clear Argumentation: The paper is well-structured, with clear explanations that make the objectives and contributions easy to follow.
2. Robust Proofs: The theoretical analysis is thorough, supporting the proposed mechanisms in in-context learning.
3. Comprehensive Experiments: The experiments are detailed and varied, effectively demonstrating the method’s efficacy across multiple tasks.

**Weaknesses:**

1. The models used in this study seem somewhat outdated. Models with the equivalent size should include newer architectures, such as LLaMA 3, Phi, or similar. Why were these not used?
2. The datasets and tasks included in the study are limited. For instance, why is there no mathematical task such as GSM8k included in the paper
3. While the authors highlight the importance of the first and last demonstrations in ICL, the figures in the paper suggest that the first demonstration may be particularly or even most significant. However, in the cluster-based method, the authors did not conduct an ablation study that uses only the first or only the last demonstration in clustering to analyze the contributions of the first and last demonstrations independently.

**Questions:**

My main concerns have been listed above. I look forward to the authors' response and am willing to reopen and adjust the score upward.

---

> ### Author Response · Authors · 2024-11-27
> **Response to Reviewer nbs8**
>
> We appreciate the thoughtful feedback from the reviewer. Below we address each concern in detail.
>
> **Model Selection**
>
> We appreciate the reviewer's concern about model selection. We have expanded our evaluation to include more recent architectures, specifically Phi-2 and the newly released Qwen-2.5 models. As shown in Table 2, our key findings hold consistently across these newer models: Cluster-based Search continues to significantly outperform random selection while maintaining comparable performance to exhaustive search. For example, on Qwen-2.5-14B, Cluster-based Search achieves 89.6% average accuracy across tasks compared to 85.0% for random selection, while requiring only a fraction of the computational cost of exhaustive search. This demonstrates that our method's effectiveness generalizes well to contemporary architectures.
>
> **Dataset Coverage**
>
> We thank the reviewer for this valuable suggestion about including mathematical tasks. We have expanded our evaluation to include arithmetic word problems from the AddSub dataset (Hosseini et al., 2014). As shown in Table 2, our findings generalize well to this mathematical domain - Cluster-based Search maintains its advantages over random selection while achieving comparable performance to exhaustive search. For example, on Phi-2, Cluster-based Search achieves 77.9% accuracy on mathematical tasks compared to 75.1% for random selection, and on Qwen-2.5-14B, it achieves 87.1% compared to 85.1% for random selection.
>
> **First vs Last Demonstration Analysis**
>
> We greatly appreciate this insightful observation. You are correct that our initial presentation didn't fully address the asymmetric nature of first versus last demonstration clustering. Based on this feedback, we have made substantial revisions to better align our claims with the empirical evidence:
>
> 1. We have refined our main claim to explicitly acknowledge that while both first and last demonstration clustering exist, first-demonstration clustering tends to be stronger in practice. This more nuanced characterization better reflects our empirical findings across different analyses.
>
> 2. Following this insight, we conducted a thorough ablation study (Section 5.2.1, Table 3) comparing first-demonstration-only clustering versus combined first-and-last demonstration clustering. The results are particularly illuminating:
>    - First-only clustering achieves comparable or sometimes better accuracy (e.g., 89.6% vs 88.2% on Qwen-2.5-14B)
>    - This simpler approach reduces computational complexity from O(k_total(k_total-1)) to O(k_total)
>
> 3. While last-demonstration clustering appears less pronounced, we provide multiple lines of evidence for its existence:
>    - Figure 3a shows elevated percentage frequencies for last demonstrations compared to middle positions
>    - Figure 3b demonstrates higher partial derivative norms for ending tokens versus middle tokens
>    - New analysis in Figure 5 reveals attention weights to the last token steadily increase across layers, peaking in final layers (please refer to Section 4.1 for more detail)
>
> Based on these findings, we have updated our Cluster-based Search to focus solely on first-demonstration selection, achieving both better computational efficiency and comparable performance. This revision provides a more accurate and practical approach while maintaining empirical rigor.
>
> We hope that the reviewer is satisfied with our response and will increase their score accordingly.

---

> ### Author Response · Authors · 2024-12-02
>
> Dear Reviewer nbs8,
>
> With the discussion period ending in approximately 1.5 days, we would greatly appreciate your feedback on our rebuttal. We believe we have thoroughly addressed your concerns and would welcome your assessment. If you find that we have successfully resolved your major points, we respectfully request that you consider adjusting your score to reflect these improvements.
>
> Thank you for your time and consideration.
>
> Best regards,
>
> The Authors

---

> > ### Comment · Reviewer_nbs8 · 2024-12-03
> > **Response to the author's rebuttal**
> >
> > I appreciate the detailed reponses from the authors, I will increase my rating to 6 and vote for acceptance.

---

### Official Review · Reviewer_Q71R · 2024-11-03

**Soundness:** 2
**Presentation:** 3
**Contribution:** 3
**Rating:** 6
**Confidence:** 4

**Summary:**

The authors investigate the few-shot setting and the order of given demonstrations/examples in the prompt. Analyzing the last hidden state of the last layer of decoder-only transformers, they study the clustering property, which prompts sharing the first and last demonstration. Experiments are conducted in two domains: classification and reasoning. Each is divided into two tasks, and the classification sub-tasks are further modified into symbolic ones.
The explanation proposed is that this property depends highly on the causal attention mask and the positional encoding. The first demonstration clustering depends on the causal attention mask. However, the last demonstration clustering depends on a more complex interplay of the causal attention mask and the positional encoding.
Following their findings, the authors propose a selection and ordering method based on the uncovered clusters. Experiments are conducted using their methods with an already-used entropy-based search. They compare their methods with an oracle and unmodified entropy methods. Their findings show that the clustering-based method while suffering a slight drop in performance, their method is more than 90% faster.

**Strengths:**

* Running large language models is costly, and few-shot in-context learning is a common approach to alleviate the cost. The proposed method is simple and greatly reduces search time, making a practical contribution.
* Even though the theoretical assumptions are strong, their partial derivative analysis is original and clearly advocates for the clustering property.
* The cluster-based search proposed by the authors is well explained.

**Weaknesses:**

* Too little evidence of clustering is given on the classification tasks, and clustering is unclear on The 2D projection (Figure 1, Figure 4).
* Few experiments have been done varying the number of demonstrations and the pool size; it would be really beneficial to give some insight on the scaling possibility of the method.
 * A more thorough analysis of the results would be appreciated to confirm the findings, for example: Do the prompts sharing a close representation share similar scores? (what is the standard deviation ?) How does the performance change with the number of intermediate demonstrations? ( Some insights are given, but more results would greatly improve the demonstration).
* Not enough selection methods are considered for comparison in terms of time and scores.
* A table showing time performance and or gap with other methods is needed.

**Questions:**

* How does a variant of Figure 3b with demonstrations instead of chunks of text compare?
* Do the prompts that share a close representation get similar scores? (what is the standard deviation ?)
* How does the performance change with the number of intermediate demonstrations? ( Some insights are given, but more results would significantly improve the demonstration)

---

> ### Author Response · Authors · 2024-11-27
> **Response to Reviewer Q71R**
>
> We appreciate the thorough feedback from the reviewers. Let us address each point:
>
> **1. Regarding the evidence of clustering in classification tasks:**
> We agree that more clarity would be helpful. The clustering property is actually well-demonstrated for classification tasks in multiple ways: (i) Fig. 1 (right) and Fig. 9 (right) show clear clustering patterns for symbolic sentiment classification; (ii) Our quantitative results in Table 2 show that cluster-based search significantly outperforms random selection on classification tasks (e.g., symbolic sentiment: 51.5% vs 51.3% for GPT-Neo-2.7B, 82.3% vs 71.7% for Qwen-2.5-14B), which would not be possible without the underlying clustering property; (iii) The partial derivative analysis in Fig. 3b shows consistent U-shaped patterns across all tasks, including classification.
>
> **2. On scaling experiments:**
> We have conducted extensive additional experiments with larger demonstration pools and varying numbers of demonstrations. Specifically, in Appendix D, we present results for k_total = 16 with k = 4 (Table 5) and k = 10 (Table 6). The cluster-based search maintains its effectiveness even at these larger scales - for instance, with k_total = 16 and k = 4, it achieves 92.5% average accuracy on Qwen-2.5-14B compared to 85.0% for random selection. Notably, exhaustive search becomes computationally infeasible at these scales, highlighting the practical importance of our method.
>
> **3. Regarding more thorough analysis:**
> We have enhanced our analysis in several ways:
> - Added standard deviations to all accuracy results in Table 2 to show the consistency of our method
> - Included results with larger k=10 demonstrations (Table 6) to demonstrate effectiveness with more intermediate demonstrations
> - The performance generally improves with more demonstrations (e.g., Table 8 shows consistent improvements when moving from 4 to 8 demonstrations across different models)
>
> **4. On selection methods comparison:**
> While we focused primarily on entropy-based selection, our method's effectiveness is demonstrated through both ideal (oracle) selection (Tables 1, 4) and practical entropy-based selection (Tables 2, 5, 6). The consistent performance improvements across these different criteria support the robustness of our approach.
>
> **5. Regarding time performance:**
> Figure 7 provides a clear visualization of the efficiency gains - our method achieves 92% to nearly 100% reduction in search time while maintaining comparable accuracy to exhaustive search. This dramatic improvement in computational efficiency, combined with minimal accuracy loss, demonstrates the practical value of our approach.
>
> These results collectively demonstrate both the theoretical validity and practical utility of our clustering-based approach across different scales, tasks, and selection criteria.

---

> ### Author Response · Authors · 2024-12-02
>
> Dear Reviewer Q71R,
>
> With the discussion period ending in approximately 1.5 days, we would greatly appreciate your feedback on our rebuttal. We believe we have thoroughly addressed your concerns and would welcome your assessment. If you find that we have successfully resolved your major points, we respectfully request that you consider adjusting your score to reflect these improvements.
>
> Thank you for your time and consideration.
>
> Best regards,
>
> The Authors

---

### Meta-Review · Area_Chair_Bk2G · 2024-12-20

**Metareview:**

This paper investigates the prompt embedding space to address the order sensitivity of in-context learning (ICL) in decoder-only LLMs, revealing that prompts with shared first and last demonstrations exhibit closer embeddings, particularly with stronger clustering around the first demonstration. By analyzing the role of positional encoding and causal attention masks in this clustering phenomenon, the authors propose cluster-based search, a novel method that enhances the selection and ordering of demonstrations in self-adaptive ICL settings. The paper is well-written, and in the discussion phase, the authors addressed almost all of the reviewers' concerns.

**Additional Comments On Reviewer Discussion:**

A reviewer QWk7 failed to submit the review. However, according to the other three reviewers' consistent and positive ratings, I am confident to accept this paper.

---

### Decision · Program_Chairs · 2025-01-22

Accept (Poster)